**Data Availability Statement:** All relevant data are within the paper and its Supporting Information files.

**Funding:** This work was supported by BiomeHub Biotechnologies. The funder provided support in

# Swab pooling: A new method for large-scale RT-qPCR screening of SARS-CoV-2 avoiding sample dilution

Ana Paula Christoff[1ᵒ], Giuliano Netto Flores Cruz[1ᵒ], Aline Fernanda Rodrigues Sereia[1], Dellyana Rodrigues Boberg[1], Daniela Carolina de Bastiani[1], Laís Eiko Yamanaka[1], Gislaine Fongaro[2], Patrícia Hermes Stoco[3], Maria Luiza Bazzo[4], Edmundo Carlos Grisard[3], Camila Hernandes[5], Luiz Felipe Valter de Oliveira[1]*

1 BiomeHub Biotechnologies, Florianópolis-SC, Brazil, 2 Laboratório de Virologia Aplicada, Departamento de Microbiologia, Imunologia e Parasitologia, Universidade Federal de Santa Catarina (UFSC), Florianópolis-SC, Brazil, 3 Laboratório de Protozoologia, Departamento de Microbiologia, Imunologia e Parasitologia, Universidade Federal de Santa Catarina (UFSC), Florianópolis-SC, Brazil, 4 Laboratório de Biologia Molecular, Microbiologia e Sorologia–LBMMS, Universidade Federal de Santa Catarina (UFSC), Florianópolis-SC, Brazil, 5 Hospital Israelita Albert Einstein (HIAE), São Paulo, SP, Brazil

ᵒ These authors contributed equally to this work.
* felipe@lfelipedeoliveira.com

## Abstract

To minimize sample dilution effect on SARS-CoV-2 pool testing, we assessed analytical and diagnostic performance of a new methodology, namely swab pooling. In this method, swabs are pooled at the time of collection, as opposed to pooling of equal volumes from individually collected samples. Paired analysis of pooled and individual samples from 613 patients revealed 94 positive individuals. Having individual testing as reference, no false-positives or false-negatives were observed for swab pooling. In additional 18,922 patients screened with swab pooling (1,344 pools), mean Cq differences between individual and pool samples ranged from 0.1 (Cr.I. -0.98 to 1.17) to 2.09 (Cr.I. 1.24 to 2.94). Overall, 19,535 asymptomatic patients were screened using 4,400 RT-qPCR assays. This corresponds to an increase of 4.4 times in laboratory capacity and a reduction of 77% in required tests. Therefore, swab pooling represents a major alternative for reliable and large-scale screening of SARS-CoV-2 in low prevalence populations.

## Introduction

The COVID-19 pandemic, caused by the severe acute respiratory syndrome coronavirus 2 (SARS-CoV-2), has dramatically impacted public health worldwide in the year of 2020 [1, 2]. Rapid identification and isolation of infected individuals is essential, but this can be particularly challenging given the infectious potential of both asymptomatic and pre-symptomatic cases [3, 4]. In this scenario, massive population SARS-CoV-2 testing is an urgent need to allow the isolation of infected individuals and, ultimately, the pandemic control.

the form of salaries for authors [APC, GNFC, AFR, DRB, DCB, LEY and LFVO], and also helped in the sample collection process, but did not have any additional role in the study design, data collection and analysis, decision to publish, or preparation of the manuscript. The specific roles of these authors are articulated in the 'author contributions' section

**Competing interests:** APC, GNFC, AFR, DRB, DCB, LEY and LFVO from BiomeHub are currently full-time employees of this company. This does not alter our adherence to PLOS ONE policies on sharing data and materials. All other authors declare no conflict of interest.

The most sensitive and recommended test for SARS-CoV-2 is based on the RT-qPCR method, which detects an active infection through the identification of the viral RNA in naso-pharyngeal samples [5–8]. However, several limitations have hampered large-scale population screenings using RT-qPCR, mainly related to the worldwide shortage of supplies and their relatively high cost. To overcome these limitations and scale-up testing capability, some research groups have proposed pooling samples for testing, in which several individuals are simultaneously analyzed using a single test [9–15].

In the many ways that pool testing was proposed so far, all of them are based on individual sample mixing by the laboratory (sample pooling). This procedure involves substantial sample manipulation, leading to operational challenges and, more importantly, to substantial dilution of viral RNA present in any of the pool samples. Such a dilution effect directly impacts the analytical sensitivity of the RT-qPCR assay, potentially leading to reduced diagnostic sensitivity [9]. Here, we describe a pooling procedure in which nasopharyngeal swabs are pooled together at the time of sample collection (swab pooling), decreasing laboratory manipulation and minimizing dilution of the viral RNA present in the sample.

## Methods

### Study design

A retrospective study was performed using de-identified results from nasopharyngeal samples subjected to RT-qPCR-based SARS-CoV-2 testing from May 5th to July 31st, 2020. Sample collection was performed focusing on low-prevalence asymptomatic or presymptomatic COVID-19 populations. Initially, a validation set consisting of 45 pool samples and all their 613 corresponding individual samples were analyzed in parallel to assess correspondence between individual and pool qualitative results. Further, 18,922 additional individuals were tested using swab pooling, corresponding to 1,344 pools and among these, only positive pools had their respective individual samples tested. Individuals from negative pools were considered negative for SARS-CoV-2 RNA detection. Comparison of paired cycle quantification (Cq) values from all positive samples obtained was carried out to assess potential quantitative biases due to swab pooling. This study was approved by the Hospital Israelita Albert Einstein Ethics Committee (number 36371220.6.0000.0071). The patient informed consent was waived off by the ethics committee as the research was performed on de-identified, anonymized samples.

### Specimen collection and swab pooling for SARS-CoV-2 screening

Nasopharyngeal samples were collected by trained healthcare professionals using nylon flocked swab, stored in tubes containing sterile saline solution and submitted to laboratory processing within a maximum of 48h after sample collection.

For the swab pooling method, two swabs were collected from the same individual (Fig 1). The first swab, collected through one nostril, was stored in an individual tube containing 3 mL of saline solution. A second swab, collected from the second nostril, was stored in a pool tube containing 5 mL of saline solution. As a general rule, each pool tube was allowed to contain up to 16 swabs, from 16 different patients, collected apart within a maximum of 1h. In this way, the pooling of swabs is performed at the time of sample collection, dismissing further manipulation, mixing and dilution of the samples by the laboratory. When a given pool tested positive, all the corresponding individual samples were also tested to identify the infected patients. If a pool yielded a negative result, all individuals within that pool were considered negative for SARS-CoV-2 RNA detection.

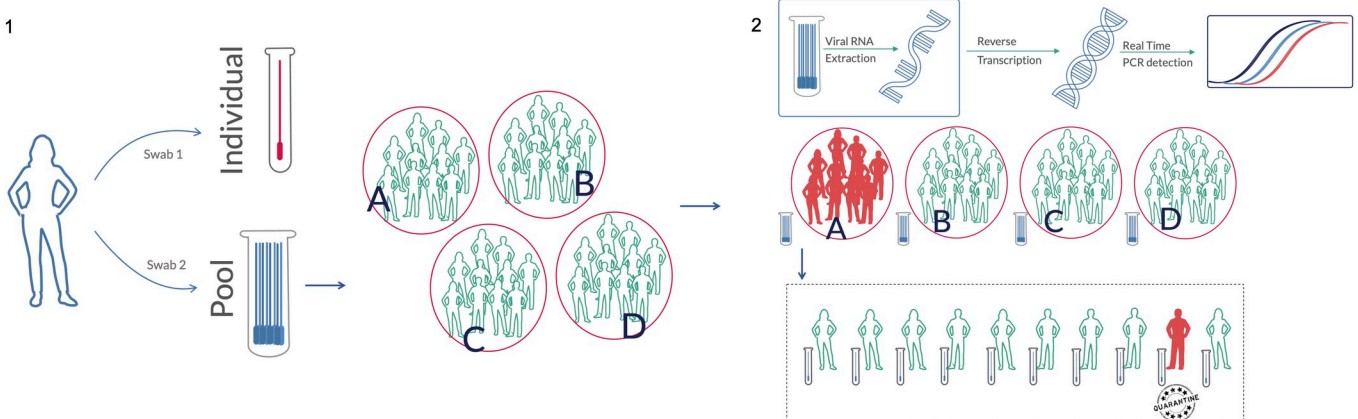

**Fig 1. Molecular screening for SARS-CoV-2 through swab pooling method.** 1) From each individual, two swabs are collected. One nasopharynx swab is collected from one nostril and stored in an individual 3 mL tube. Then, another swab is collected through the other nostril and stored in a 5 mL tube containing up to 15 other individuals (pool tube). 2) In the laboratory, the RNA from pooled swabs is extracted, and SARS-CoV-2 detection is performed using RT-qPCR. If a given tested pool presents a positive result, all corresponding individual samples are then processed to identify infected patients.

### SARS-CoV-2 RT-qPCR detection

RNA isolation was performed from nasopharyngeal samples using guanidine thiocyanate lysis solution followed by magnetic beads capture and purification (BiomeHub, Brazil). Samples were eluted in 40 μL of RNAse-free water. RNA reverse transcription (RT) was performed using SupesScript[TM] IV (Invitrogen, USA) and random hexamers, according to the manufacturer instructions.

Real-Time PCR detection of SARS-CoV-2 was performed using the following genetic markers: a region of the gene encoding the viral envelope protein (*E*) with P1 probe and the RNA-dependent RNA polymerase gene (*RdRp*) with P2 probe for discriminatory assay. Primers, probes and protocols are described in the Charité-Berlin publication [8, 16]. Also, data from the detection of the surface glycoprotein gene (*S*) using SYBR Green intercalating fluorophore as previously described [17] were included. Amplifications were performed in 7500 Fast, QuantStudio 6 Pro Real Time PCR (Applied Biosystems, USA), or in a CFX 384 (BioRad, USA). Cycle quantification values from the RT-qPCR amplifications were used for data analysis. In order to consider a sample positive for the presence of SARS-CoV-2 RNA, we proceeded as follows: for pool samples, detecting at least one gene with Cq value lower than 40 was enough for pool opening and subsequent individual testing; for individual samples, detecting both tested genes with Cq lower than 40 was required to consider it positive for SARS-CoV-2.

### Statistical analysis

All statistical analyses were performed using R statistical software (v. 3.6.3) [18]. Data wrangling and visualization were performed using the tidyverse package suite (v. 1.3.0) [19]. Modeling was performed using the brms R package (v. 2.12.0) and the Stan probabilistic programming language (v. 2.19.1) [20, 21]. Additional R packages included ggpubr (v. 0.2.5), RColorBrewer (v. 1.1.2), binom (v. 1.1.1), and patchwork (v. 0.0.1) [22–25].

Concordance between pool and individual tests was determined by considering their corresponding qualitative results, *i.e.*, a test was considered concordant if the individual result matched the result from its corresponding pool. Among positive tests, we quantified the mean Cq difference between individual samples and their corresponding pools. We employed a

Bayesian hierarchical model with patient-specific intercepts as follows:

$$Cq_i \sim N(\mu_i, \ \sigma^2)$$
$$\mu_i = \alpha + \alpha_{patient[i]} + \beta_{pool} * Pool_i + \beta_E * E_i + \beta_{RdRp} * RdRp_i$$
$$+ \beta_{Pool:E} * Pool_i * E_i + \beta_{Pool:RdRp} * Pool_i * RdRp_i$$
$$\alpha_{patient} \sim N(0, \sigma_\alpha)$$
$$\alpha \sim N(25, \ 3) \tag{1}$$
$$\beta. \sim N(0, \ 5)$$
$$\sigma_\alpha \sim Exponential(1)$$
$$\sigma \sim Exponential(1)$$

where $Pool_i$ is an indicator variable which equals 1 when the $i^{th}$ observation is from a pool sample and 0 otherwise. The $E_i$ and $RdRp_i$ variables adjust for variation in the genetic marker used for the RT-qPCR assay. In these settings, the population-level intercept $\alpha$ represents the average Cq value for an individual test using the $S$ gene, whereas the patient-specific intercept $\alpha_{patient[i]}$ accounts for patient-to-patient variability. The $\beta$ coefficients allow quantification of mean Cq differences between individual and pool tests across different genetic markers. We set weakly informative priors for all parameters. Results were reported as posterior means and 95% credible intervals.

For the sample dilution in swab pooling experiment, the same model as in (1) was employed, except that varying intercepts varied with inoculating samples instead of with patients; also, only $E$ and $RdRp$ genes were used. Credible intervals for proportions were obtained using a Beta(1, 1) prior for the binomial likelihood. Observed correlations were reported as Spearman's rank correlation as well as Pearson's correlation coefficient.

## Results

### Sample dilution in swab pooling—proof of concept

In a laboratory experiment, 16 positive nasopharyngeal samples were selected as inoculating samples to be mixed in equal volumes into 16 negative pool samples as well as into 16 negative individual samples, according to a dilution factor of 1.67. This dilution factor corresponds to volumes between samples collected in swab pooling tubes (with 5 mL saline solution) and samples collected in individual tubes (with 3 mL saline solution). For this paired experiment, we observed a mean Cq difference between pool and individual samples of 0.42 Cq (95% Cr.I. -0.22 to 1.09) for the $E$ gene and 0.6 Cq (95% Cr.I. -0.05 to 1.24) for the $RdRP$ gene (Fig 2A).

In Fig 2B we show the expected slopes ($\Delta$Cq) in RT-qPCR amplifications with variable amplification efficiencies and considering different dilution scenarios. In sample pooling, the dilution factor is equal to the number of individual samples within a pool, yielding expected mean Cq differences of at least 3.32 Cq and as high as 7.37 Cq depending on the number of pooled samples and the amplification efficiency. For swab pooling, on the other hand, the dilution factor is kept fixed at 1.67 so that the expected variation due to dilution alone is constrained between 0.73 and 1.08 Cq.

### Paired analysis of pool and individual tests

To investigate any loss of diagnostic sensitivity due to swab pooling, we analyzed individual and pool samples from 613 patients regardless of their pool results (*i.e.*, positive and negative pools). All the individual and pool samples were analyzed in parallel resulting in 94 positive

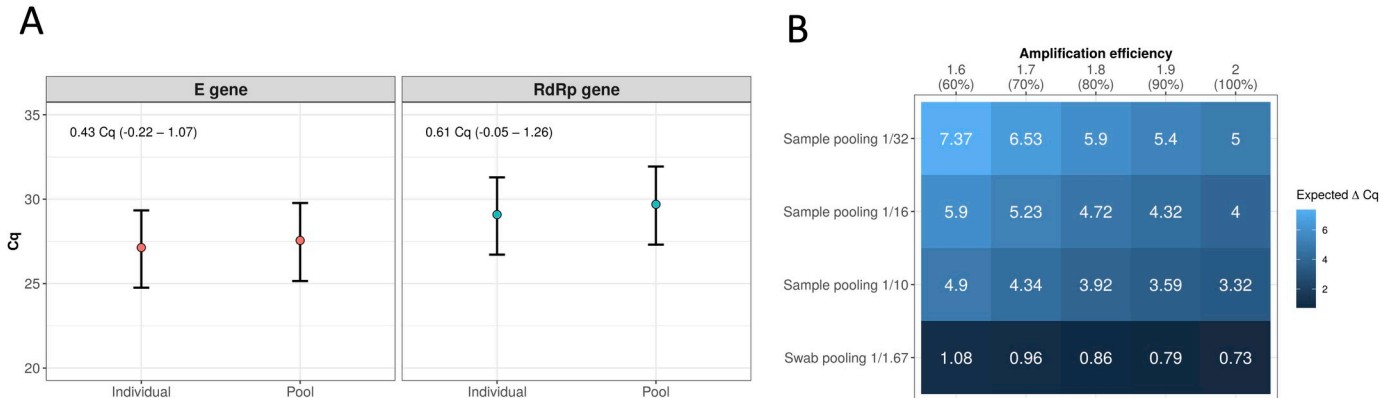

**Fig 2. Swab pooling and dilution effects.** (A) In a laboratory controlled experiment, 16 positive samples were inoculated in negative individual and pool samples maintaining the dilution factor of 1.67 between individual (3 mL) and swab pooling (5 mL) samples. Mean Cq differences between individual and pool samples estimated for *E* and *RdRp* genes along with 95% credibility intervals are shown in the top-left corner of the graphs. (B) Expected Cq variations (ΔCq) for sample pooling methods with 10, 16 or 32 samples compared to swab pooling in different amplification efficiencies. Expected ΔCq was calculated using Efficiency$^{slope}$ = dilution factor [29–31]. In swab pooling, the number of samples are not related to the dilution factor.

individual tests and 20 positive pools (Fig 3A). Among the 20 positive pools, at least one individual sample in each pool tested positive for SARS-CoV-2. Positive patients per pool varied from 1 to 11 (Fig 3B). We observed no clear evidence of correlation between the pool Cq values and the number of positive samples within each pool (Fig 3C). Paired comparisons of the pool and their respective individual Cq values can be visualized in Fig 3D. Further analysis of Cq variation is performed in the next section.

Qualitatively, we did not observe any positive individual test paired with a negative pool, *i. e.*, no false-negatives due to swab pooling. In fact, we observed complete agreement (100%) between qualitative results from the pool and individual paired samples. Hence, we employed a simple beta-binomial model with flat priors on performance estimates that would otherwise reach 100%. Having individual testing as a reference, the current data supports a sensitivity of 99% (95% Cr.I. 96.9% to 100%) and a specificity of 99.8% (95% Cr.I. 99.4% to 100%) for the swab pooling procedure, indicating evidence of strong similarity in diagnostic performance.

## Large-scale screening for SARS-CoV-2 using swab pooling

To investigate any biases in quantitative results, we included data from additional 1,344 pools and their respective individual tests. In total, 19,535 patients (1,389 pools) were screened using the swab pooling method herein described. Considering all combined results, we observed 246 positive patients for SARS-CoV-2 distributed in 163 pools, resulting in a positivity rate of 1.26%. For 12 pools (0.86%), amplification of both *E* and *RdRp* genes was detected but no associated positive individual sample was identified. In such cases, a new sample collection was requested by the laboratory.

Among the 163 positive pools, 100 (61.3%) contained exactly 16 pooled swabs (Fig 4A). Also, 104 pools (63.8%) corresponded to exactly one positive individual test each. Over 81% of positive pools presented at most 3 correspondent positive individual tests. From all our data, four pools showed Cq values above 40 for one gene, but below that threshold for the other gene (41.06, 40.58, 40.58 for the *E* gene against 35.32, 37.52, 36.39 for the *RdRp* gene, and 44.79 for the *RdRp* gene against 34.83 for the *E* gene). This is in accordance with our requirements for opening pools tested during validation. We observed increased Cq results for three individual samples: 41.93, 41.61, 40.99 for the *E* gene against 34.58, 35.74 and 37.27 for the *RdRp*

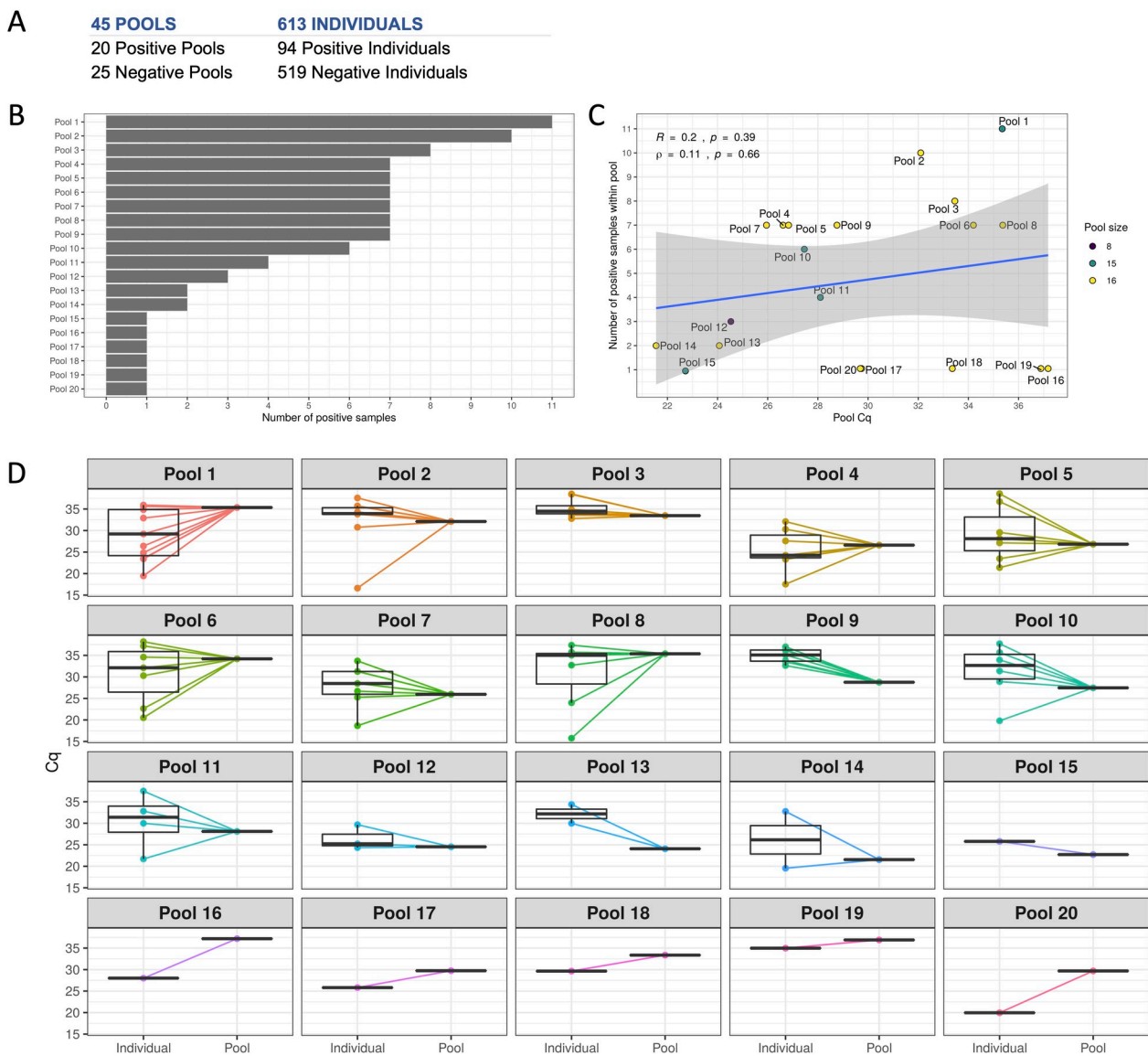

**Fig 3. Paired analysis of pool and individual tests.** (A) Results for samples analyzed in parallel as pools and as individual tests. (B) Total number of positive samples for each positive pool. (C) Correlation between the number of positive samples within a given pool and the corresponding pool Cq (*R*: Pearson's corr. coefficient; ρ: Spearman's rank corr. coefficient). Points were colored by the total number of individuals within the pool (pool size). (D) Cycle quantification values for positive pools and corresponding positive individual samples.

gene. However, it was not possible to repeat sample collection and analyzing the amplification curves profiles, these three patients were considered positive and, hence, their Cq values were kept in the analysis.

Correlation between Cq values from individual tests and their corresponding pools was strongest for pools associated with one or two positive samples, seemingly diminishing with the increase in the number of positive samples within the pools (Fig 4B). To estimate the Cq variation due to swab pooling, we assigned to each patient the Cq value from their individual test and the Cq value from their respective pool. Using a hierarchical model with patient-specific intercepts, we estimated the mean Cq difference between individual tests and their corresponding pools for each genetic marker (Fig 4C). For the *S* gene (94 patients), the mean Cq

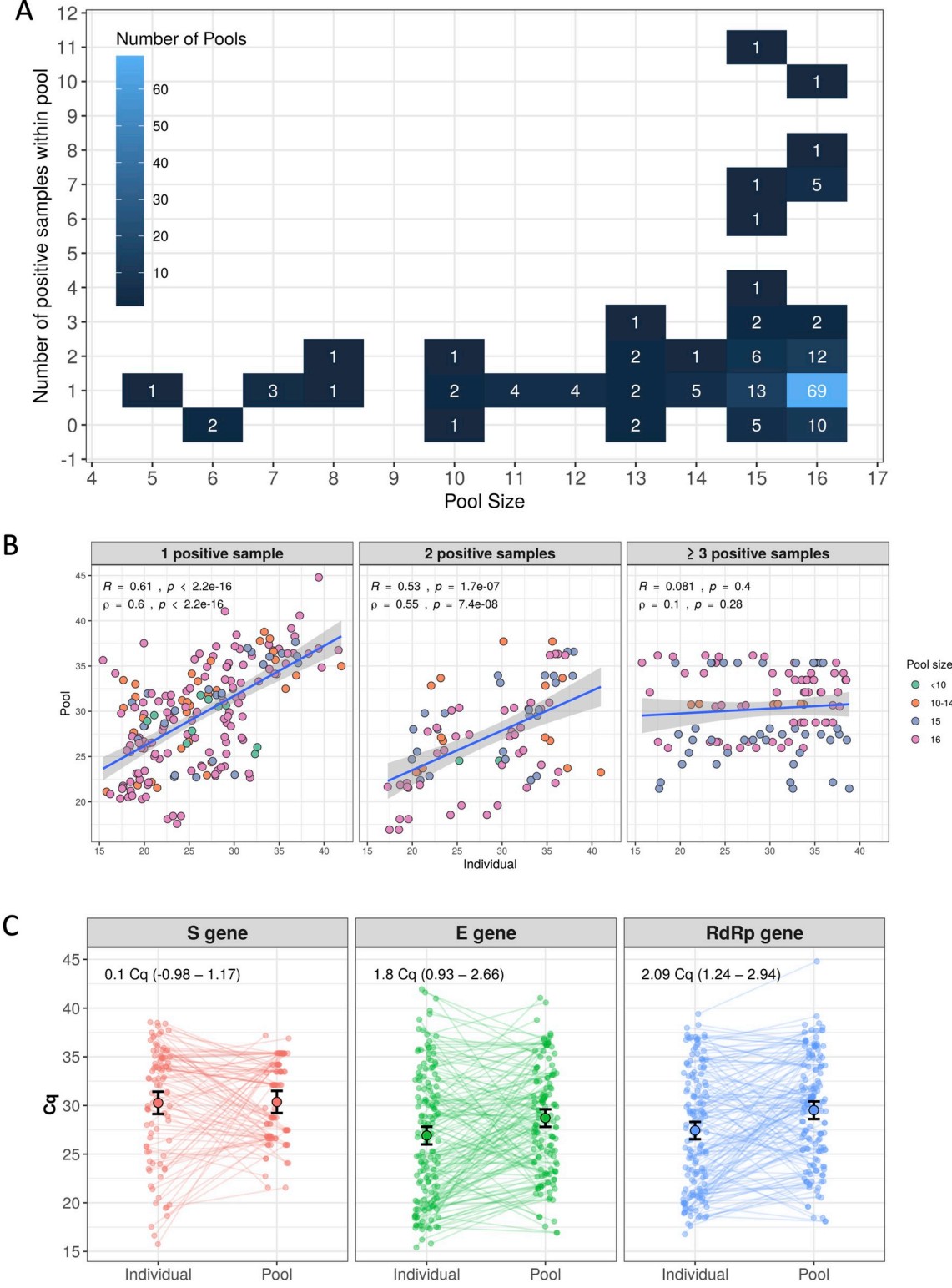

**Fig 4. Large-scale molecular screening of SARS-CoV-2 with swab pooling.** (A) Number of positive pools related to the original pool size and the number of positive samples within the pool. (B) Correlation between Cq values from pools and their respective individual samples stratified by the number of positive samples within the pool (1, 2, and 3 or more positive individual samples). Point color represents the pool size (from less than 10 to 16 individuals). Cq values for the three marker genes tested were included. Correlation coefficients are presented in the figure (*R*: Pearson's corr. coefficient; ρ: Spearman's rank corr. coefficient). (C) Cq values from

individual tests and corresponding pools were assigned to each patient. A hierarchical model with patient-specific intercepts was used to estimate mean variations between individual and pool samples across varying genetic markers. Estimates for mean Cq differences along with 95% credible intervals are shown in the top-left corner of each graph.

difference was estimated to be 0.1 Cq (95% Cr.I. -0.98 to 1.17). Differences for the *E* and *RdRp* genes (152 patients) were estimated to be 1.8 Cq (95% Cr.I. 0.93 to 2.66) and 2.09 Cq (95% Cr. I. 1.24 to 2.94), respectively.

## Discussion

Extensive SARS-CoV-2 testing is essential for monitoring human infection and investigating viral spread. Pool testing has gained importance to fight the COVID-19 pandemic, as challenges involving cost and logistics are at the core of shared struggles to promote large-scale screenings worldwide [10]. Traditional pooling methods proposed in other studies rely on the combination of multiple individual samples prior to RNA extraction or RT-qPCR, leading to a sample dilution factor directly related to the number of samples in the pool [9, 11–15]. This dilution effect has been of major concern over the diagnostic performance of pool testing procedures [26]. Here, we report a pooling strategy that readily minimizes such dilution effect and enables large-scale screening for SARS-CoV-2 with negative results generally 24h after sample collection and positive results in a maximum of 48-72h after sample collection.

Assessing data from our 19,535 screened patients, swab pooling and individual testing showed hardly distinguishable performances both qualitatively and quantitatively. With complete agreement between paired qualitative results from 613 patients, the presented data indicates evidence of strong similarity in diagnostic sensitivity and specificity. We did not observe a clear correlation between pool Cq values and number of positive individuals within the pools, as previously suggested considering other pooling methods [14]. Also, the correlation between Cq values from individual tests and their corresponding pools seems to be stronger for pools with no more than two positive individuals. This is mainly reflecting the potentially wide range of individual Cq values for samples composing a single pool.

Although we do use a larger volume for swab pooling (5 mL of saline solution versus 3 mL in individual tubes), the corresponding dilution factor of 1.67 will lead to an expected mean increase of 1.08 Cq even under sub-optimal amplification efficiencies. In a laboratory-controlled experiment, we did not detect clear differences due to dilution alone, with point estimates from 0.43 to 0.61 Cq. In practice, observational data from 246 positive patients generated point estimates of mean Cq differences between individual tests and their corresponding pools ranging from 0.1 to 2.09 Cq. While such values are hardly significant in terms of analytical sensitivity, the expected counterparts for traditional pooling would range from 3.3 to 5 Cq under optimal amplification conditions. This range corresponds to dilution factors from 10 to 32, when equivolumetric pools from 10 to 32 samples, respectively, are formed post-collection by the laboratory as traditionally proposed [9, 11, 13, 15]. In a worst-case scenario for swab pooling, a mean Cq difference of 2.94 Cq (*RdRp* gene, upper limit of 95% credible interval) would still be considerably lower than the expected differences for sample pooling with 10 samples and perfect amplification efficiency. Nonetheless, there is always a limitation towards samples with Cq's higher than 35, in which case mean differences as small as 1 Cq could still result in false-negative tests regardless of the pooling strategy.

In practice, the major difference between swab pooling and traditional pooling methods regards sample collection: while in swab pooling we combine multiple swabs in the same tube at the time of sample collection, traditional strategies pool equal volumes from individually collected samples after sample collection, in the laboratory. Besides the greater dilution factor,

to perform sample pooling accordingly with traditional methods adds complexity to laboratory operations and may lead to increased workload to already saturated laboratory facilities. Traditional pooling requires significant sample manipulation to perform aliquots and grouping of samples with an increased risk of contamination and even possible sample exchange during the laborious pooling process. This also adds significant time to sample processing and releasing results.

On the other hand, collecting two swabs from the same patient can be operationally trivial. While one swab goes into the pooling tube, the other one will only be processed by the laboratory if the pool tests positive, this facilitates the sample handling by the laboratory and decreases the time and complexity of performing the traditional pooling. In cases where it's not possible to collect two swabs from the same patient, the sample should not be included in the swab pooling tube and analyzed as individual diagnostics instead. A critical step, this sample collection process can still represent an important limitation of swab pooling as it can cause variation between pooled and individual swabs. In this study, we detected 12 pools with positive results but no positive associated individual test. Of these, 8 pools were associated with two specific collection events (4 pools collected each day). Thus, it is likely that such inconsistencies are attributable to the sample collection process. Still, these cases represented 0.86% of all 1,389 tested pools. Notably, the proper training of sample collection staff represents a cheaper and easier-to-implement alternative to increased laboratory complexity. Any laboratory capable of routine processing of diagnostic samples for SARS-CoV-2 can also perform swab pool analysis using the same detection methods and infrastructure already in use.

Using swab pooling during sample collection, laboratories in which traditional pooling is currently unfeasible become readily able to contribute to large-scale screenings. Swab pooling, therefore, represents a gain in operational performance for reliable testing of SARS-CoV-2 at scale. As it is well-known, however, any pooling strategy only boosts testing capability for low positivity rates [27]. Swab pooling does not address this matter and is, therefore, suitable for screening populations a with low expected prevalence of COVID-19.

The data in the present study comes from the application of swab pooling in asymptomatic or presymptomatic populations, yielding a 1.26% positivity rate. The proposed method was used with pools containing a majority of 16 individuals, but the optimum pool size can be determined by each laboratory during internal validation. Pools with 8, 10, 16, or even 32 swabs may be desirable depending on local epidemiological status and target populations. Upon validation, swab pooling may be applied to any reasonable pool size traditionally proposed to optimize testing scale. Here, over 19,500 patients were screened using approximately 4,400 RT-qPCR assays, corresponding to an increase of 4.4 times in laboratory capacity and a reduction of 77% in the total of required tests.

Finally, identification of infected patients is essential to contain the spread of SARS-CoV-2. This has been hampered by the fact that several people carrying the virus remain asymptomatic or presymptomatic [4, 28]. Thus, massive and sensitive testing of asymptomatic and presymptomatic individuals is of utmost importance to fight the COVID-19 pandemic, especially at the moment in which the world attempts to resume economic and social activities.

## Conclusion

Pool testing is a major alternative for large-scale screening of SARS-CoV-2 in low prevalence populations. Here, we demonstrate that the swab pooling minimizes sample dilution, can be as sensitive as individual testing and reduces laboratory workload. A total of 77% of tests were saved in the screening of 19,535 asymptomatic or presymptomatic patients.

## Supporting information

**S1 Data.**

(GZ)

## Acknowledgments

We would like to thank all BiomeHub, HIAE, and UFSC staff who were involved in all stages of sample collection, laboratory processing, and results discussion. We are grateful for their countless efforts to persist in high-quality research during such difficult times for science in our country, especially during the COVID-19 pandemic.

## Author Contributions

**Conceptualization:** Ana Paula Christoff, Giuliano Netto Flores Cruz, Aline Fernanda Rodrigues Sereia, Dellyana Rodrigues Boberg, Gislaine Fongaro, Patrícia Hermes Stoco, Maria Luiza Bazzo, Edmundo Carlos Grisard, Luiz Felipe Valter de Oliveira.

**Data curation:** Ana Paula Christoff, Giuliano Netto Flores Cruz, Edmundo Carlos Grisard, Luiz Felipe Valter de Oliveira.

**Formal analysis:** Giuliano Netto Flores Cruz, Dellyana Rodrigues Boberg, Daniela Carolina de Bastiani, Laís Eiko Yamanaka, Luiz Felipe Valter de Oliveira.

**Funding acquisition:** Luiz Felipe Valter de Oliveira.

**Investigation:** Giuliano Netto Flores Cruz, Aline Fernanda Rodrigues Sereia, Dellyana Rodrigues Boberg, Luiz Felipe Valter de Oliveira.

**Methodology:** Ana Paula Christoff, Aline Fernanda Rodrigues Sereia, Dellyana Rodrigues Boberg, Edmundo Carlos Grisard, Luiz Felipe Valter de Oliveira.

**Project administration:** Dellyana Rodrigues Boberg, Luiz Felipe Valter de Oliveira.

**Resources:** Gislaine Fongaro, Patrícia Hermes Stoco, Maria Luiza Bazzo, Camila Hernandes, Luiz Felipe Valter de Oliveira.

**Software:** Giuliano Netto Flores Cruz.

**Validation:** Ana Paula Christoff, Giuliano Netto Flores Cruz, Luiz Felipe Valter de Oliveira.

**Writing – original draft:** Ana Paula Christoff, Giuliano Netto Flores Cruz.

**Writing – review & editing:** Ana Paula Christoff, Giuliano Netto Flores Cruz, Aline Fernanda Rodrigues Sereia, Dellyana Rodrigues Boberg, Gislaine Fongaro, Patrícia Hermes Stoco, Maria Luiza Bazzo, Edmundo Carlos Grisard, Camila Hernandes, Luiz Felipe Valter de Oliveira.

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
