## [Decision Letter · Decision Letter 0]

15 Dec 2020

PONE-D-20-32256

Swab pooling: a new method for large-scale RT-qPCR screening of SARS-CoV-2 avoiding sample dilution

PLOS ONE

Dear Dr. de Oliveira,

Thank you for submitting your manuscript to PLOS ONE. After careful consideration, we feel that it has merit but does not fully meet PLOS ONE’s publication criteria as it currently stands. Therefore, we invite you to submit a revised version of the manuscript that addresses the points raised during the review process.

In agreement with the referee ii have major critiques and concerns on this manuscript:

1- the time line of the overall process from swab collection, swab pooling, analysis  by RTPCR and availability of the data; at the same time there is no parallel comparison between the method proposed and the standard one or that using antigen detection.

2- Ct values of 40 or above seems much to high; details should be given regarding the COV2 ODNs used in this study, and why Ct's of 40 have been used.

We look forward to receiving your revised manuscript.

Kind regards,

Jean-Luc EPH Darlix, MG, Ph.D.

Academic Editor

PLOS ONE

Journal Requirements:

'This work was supported by BiomeHub Biotechnologies.'

We note that one or more of the authors have an affiliation to the commercial funders of this research study,  BiomeHub Biotechnologies.

Reviewers' comments:

Reviewer's Responses to Questions

**Comments to the Author**

1. Is the manuscript technically sound, and do the data support the conclusions?

Reviewer #1: Yes

2. Has the statistical analysis been performed appropriately and rigorously? 

Reviewer #1: Yes

3. Have the authors made all data underlying the findings in their manuscript fully available?

Reviewer #1: No

4. Is the manuscript presented in an intelligible fashion and written in standard English?

Reviewer #1: Yes

5. Review Comments to the Author

Reviewer #1: In the current paper, Christoff et al. report a large-scale swab-pooling method for SARS-CoV-2 detection. The overall idea is that this method avoids sample dilution and is based on pooling nasopharyngeal swabs from multiple patients thus decreasing laboratory manipulation.

The manuscript is well written and experiments seem to be correctly performed. However, since the authors argue on the high reliability of the assay, decreased lab manipulation, and an overall reduction of 77% of the PCR tests, I have the following concerns about its implementation in practice which need to be addressed:

1. The sampling time and sample handling are increased compared to the “classical” test since 1 tube will be systematically collected apart of the “pool tube” and 2 swabs will be needed per patient instead of 1. Moreover, in practice it is not always possible to perform sampling from both nostrils especially in children.

2. Although the swab pooling could decrease the number qRT-PCR reactions, this would not be valid for populations where the infectivity rate is high and consequently, the percentage of positive PCR samples (10-30% or even more). In fact, in the current study, the positivity rate of the tests is 1,26% which is quite low compared to the actual situation in many countries where high numbers such as 40% of test positivity rates could be observed. In such cases, the number of qRT-PCR reactions will even increase.

3. One of the main challenges of SARS-CoV-2 diagnostics is its rapidity in order to isolate/quarantine the positive patients. The proposed method can only discard at first the negative ones, but further analyses are needed to confirm the true positive cases which may take 2 more days from sampling, a period during which the patients are highly contagious.

4. No comparison of the proposed method with the SARS-CoV-2 antigen tests is made. The latter are currently becoming quite popular especially for large scale screens of potentially infected populations and asymptomatic patients and can reveal infection in less than 30 min.

5. I have reserves about considering qPCR cut off values of 40 qC and above as positive. What is the PCR program and the primers that were used? This section needs to be more detailed (not only references) since the whole study is based on qPCR. It would be worthy to include as a supplementary figure a sample image of PCR plots for the different SARS-CoV-2 genes.

6. Figure 5 could be included in Figure 4.

6. PLOS authors have the option to publish the peer review history of their article (what does this mean?). If published, this will include your full peer review and any attached files.

Reviewer #1: No

---

## [Author Response · Author response to Decision Letter 0]

22 Dec 2020

Response to reviewers

PONE-D-20-32256

Swab pooling: a new method for large-scale RT-qPCR screening of SARS-CoV-2 avoiding sample dilution

PLOS ONE

Dear Dr. de Oliveira,

Thank you for submitting your manuscript to PLOS ONE. After careful consideration, we feel that it has merit but does not fully meet PLOS ONE’s publication criteria as it currently stands. Therefore, we invite you to submit a revised version of the manuscript that addresses the points raised during the review process.

In agreement with the referee ii have major critiques and concerns on this manuscript:

1- the time line of the overall process from swab collection, swab pooling, analysis by RTPCR and availability of the data; at the same time there is no parallel comparison between the method proposed and the standard one or that using antigen detection.

2- Ct values of 40 or above seems much to high; details should be given regarding the COV2 ODNs used in this study, and why Ct's of 40 have been used.

Response: All editor's questions were also addressed within the responses to the reviewer's comments. We also included in the manuscript the timeline of sample processing considering negative and positive pools. After sample arrival at the laboratory, patients from positive pools have results released within 48-72 hours; patients from negative pools have results released within 24 hours. 

In this manuscript, the swab pooling method was compared to a standard method, the individual nasopharyngeal RT-qPCR assay, performed as recommended by WHO and regulatory agencies in most countries. Thus, our pooling method was compared to the standard RT-qPCR diagnostic method. 

Some Cq values above 40 were included (very few cases that appeared along our study) just to show all the data we had, since this manuscript is a demonstration of a new approach for SARS-CoV-2 screening. CDC and Charité-Berlin protocols, the ones used worldwide, perform 45 amplification cycles with a cutoff of 40 Cq, but as we deal with clinical samples and biological data, these extreme thresholds should be carefully analyzed. We do not intend to state that all Cqs of 40 and above should be considered positive, but we included these few cases to highlight that such things happen with real biological samples and should be evaluated along with the laboratory expertise. Several SARS-CoV-2 papers (as the links below) showed Cqs of 40, which is a very common practice for viral RNA detection in clinical samples, given the low viral load in the samples and the inherent PCR stochasticity in that range.

https://www.eurosurveillance.org/content/10.2807/1560-7917.ES.2020.25.32.2001483

https://www.nature.com/articles/s41564-020-0761-6

https://jcm.asm.org/content/58/5/e00310-20

https://jamanetwork.com/journals/jamainternalmedicine/fullarticle/2769235

https://www.nature.com/articles/s41467-020-18611-5

https://jamanetwork.com/journals/jama/fullarticle/2765837

https://academic.oup.com/cid/article/71/16/2073/5828059

https://www.ncbi.nlm.nih.gov/pmc/articles/PMC7267454/

https://www.thelancet.com/journals/lancet/article/PIIS0140-6736(20)30154-9/fulltext

Journal Requirements:

A revised cover letter version was submitted including the statements required.

All the data used in the study was made available.

The financial disclosure statement was adjusted as requested.

Reviewer #1:

3. Have the authors made all data underlying the findings in their manuscript fully available?

Response: Data and code to reproduce all the analyses included in this manuscript are provided in Supplementary information.

Reviewer #1: In the current paper, Christoff et al. report a large-scale swab-pooling method for SARS-CoV-2 detection. The overall idea is that this method avoids sample dilution and is based on pooling nasopharyngeal swabs from multiple patients thus decreasing laboratory manipulation.

The manuscript is well written and experiments seem to be correctly performed. However, since the authors argue on the high reliability of the assay, decreased lab manipulation, and an overall reduction of 77% of the PCR tests, I have the following concerns about its implementation in practice which need to be addressed:

1. The sampling time and sample handling are increased compared to the “classical” test since 1 tube will be systematically collected apart of the “pool tube” and 2 swabs will be needed per patient instead of 1. Moreover, in practice it is not always possible to perform sampling from both nostrils especially in children.

Response: We thank the reviewer for the important consideration. In fact, there can be an increase in sampling time compared to workflows that perform the collection of only one swab. However, several countries guidelines already suggest two swabs in the collection procedure, from both nostrils, from nasopharynx and oropharynx, among several other variations around the world. This minimal difference for sample collection in the swab polling method is of negligible magnitude, especially when compared to the overall decrease in average processing time per patient, i.e., screening over 20 thousand individuals would be far more time consuming with no pool testing at all. With the swab pooling method implemented here, the collection procedure last less than 2 minutes. Within the laboratory, sample handling from swab pooling is faster and safer compared to the classical sample pooling procedure, once steps of making new identifications, aliquots, and sample manipulation to make the pools are substantially laborious. These steps also require more people involved, which is a critical operational limitation. Additional sample manipulation could also lead to increased chance of contamination or even sample/pool misidentifications. In the swab pooling method, we aimed to reduce and simplify such laborious and time-consuming requirements. These considerations are detailed in lines 236-250. In the rare cases when two swabs cannot be sampled, individual testing is performed without major operational harm and processed as a single diagnostic test.

2. Although the swab pooling could decrease the number qRT-PCR reactions, this would not be valid for populations where the infectivity rate is high and consequently, the percentage of positive PCR samples (10-30% or even more). In fact, in the current study, the positivity rate of the tests is 1,26% which is quite low compared to the actual situation in many countries where high numbers such as 40% of test positivity rates could be observed. In such cases, the number of qRT-PCR reactions will even increase.

Response: We agree with the reviewer that the cost-effectiveness of swab pooling relies on limited positivity rate. The infection rate of 1,26% reflects our local scenario from May-July 2020, when the samples were collected. Noteworthy, this is a limitation inherent to the idea of pool testing in general - the scale gains are inversely proportional to the number of positive individuals in the sampled population. In fact, our methodology is flexible to accommodate varying pool sizes effortlessly with fixed dilution factor, which can be used to customize pool testing to scenarios of higher expected prevalence. In traditional pool testing, varying pool size means varying dilution factors and laboratory protocols, which may demand additional validation. These considerations are discussed in the paper (lines 259 to 264). 

3. One of the main challenges of SARS-CoV-2 diagnostics is its rapidity in order to isolate/quarantine the positive patients. The proposed method can only discard at first the negative ones, but further analyses are needed to confirm the true positive cases which may take 2 more days from sampling, a period during which the patients are highly contagious.

Response: We share the concern about waiting time for testing, a major issue almost a year after the first COVID-19 outbreak. We do not claim that swab pooling supersedes massive individual testing if this is a viable option. Yet, the demand for SARS-CoV-2 testing has reached levels far beyond the testing capacity of most countries, and massive screenings are considerably rare. Also, by saving a substantial number of RT-qPCR assays, swab pooling reduces the demand for laboratory supplies, optimizing the currently overloaded supply chain. In this scenario, swab pooling boosts testing capacity when the target population is expected to yield a low positivity rate - e.g. screening of asymptomatic individuals, in which case the people being tested are not choosing between pool testing and a faster alternative, but between pool testing and no testing at all. In this sense, the strategic advantage of pool testing is discarding negative individuals, as is the case with any screening strategy. Finally, it’s important to notice that waiting time rarely exceeds 48 hours independently of test positivity, and a negative result may be released even sooner.

4. No comparison of the proposed method with the SARS-CoV-2 antigen tests is made. The latter are currently becoming quite popular especially for large scale screens of potentially infected populations and asymptomatic patients and can reveal infection in less than 30 min.

Response: Antigen testing for SARS-CoV-2 is an important strategy that has become quite popular. However, the Diagnostic flow diagram for the detection of acute SARS-CoV-2 infection in individuals with clinical suspicion for COVID-19, recommended by the World Health Organization, is still based on nucleic acid amplification methods (Fig 1). Here, we have compared swab pooling with the standard of care (individual RT-qPCR tests) and present it as a reliable and efficient alternative to complement available strategies. Our method has allowed the screening of over 20 thousand individuals who would have been tested by neither individual RT-qPCR or antigen testing.

5. I have reserves about considering qPCR cut off values of 40 qC and above as positive. What is the PCR program and the primers that were used? This section needs to be more detailed (not only references) since the whole study is based on qPCR. It would be worthy to include as a supplementary figure a sample image of PCR plots for the different SARS-CoV-2 genes.

Response: This specific threshold value of 40 has been used before and is stated in the CDC interim guidelines (https://www.fda.gov/media/134922/download). Also, the protocol was used as described in the Charité-Berlin publication (https://doi.org/10.2807/1560-7917.ES.2020.25.3.2000045). Recently, assessment of analytical sensitivity and efficiency of SARS-CoV-2 primer–probe sets also employed a cutoff of 40 Cq for clinical samples (https://www.nature.com/articles/s41564-020-0761-6). Other studies also related the COVID-19 infectiousness with RT-PCR cycle thresholds and showed several Cq values above 40 (https://www.eurosurveillance.org/content/10.2807/1560-7917.ES.2020.25.32.2001483). In our study, we performed 45 cycles in the RT-qPCR reaction, and for conservativeness, we opened pools with 40 Cq or little higher for any of the tested genes given that lower viral loads have higher Cq values and a higher rate of false-negatives due to the RT-qPCR methodology limitations. We only had 3 pools with Cqs higher than 40 for the E gene (41.06, 40.58, 40.58) and 3 individuals (41.93, 41.61, 40.99), and one exceptional pool with Cq 44.79 for the RdRp gene as its E gene Cq was much lower than 40, which could indicate some PCR deviation. These 4 pools and 3 individuals is a really small fraction of all the data we had (163 pools and 246 individuals) and is explained in lines 186-193. We only included these data to demonstrate the full range of results obtained for the method experimental validation. The PCR program and the primers used were described in the methodology section according to their original references. The Charité-Berlin protocol is one of the most well-known and recommended protocols to be followed by WHO (lines 113-120). These experiments were performed in several RT-qPCR plates, it will be a little impractical and also uninformative to concatenate all the RFU values (fluorescence at each PCR cycle) to construct the PCR plots for all samples. All Cq values were already demonstrated in the new figure 4C comparing the Cq of individuals and their respective pools. The raw data are available in the supplementary information.

6. Figure 5 could be included in Figure 4.

Response: We thank the reviewer for the suggestion and have moved Figure 5 to Figure 4C.

---

## [Editor Report · Decision Letter 1]

14 Jan 2021

PONE-D-20-32256R1

Swab pooling: a new method for large-scale RT-qPCR screening of SARS-CoV-2 avoiding sample dilution

PLOS ONE

Dear Dr. de Oliveira,

Thank you for submitting your manuscript to PLOS ONE. After careful consideration, we feel that it has merit but does not fully meet PLOS ONE’s publication criteria as it currently stands. Therefore, we invite you to submit a revised version of the manuscript that addresses the points raised during the review process.

I still have a number of critiques: (i) the reason of using amplification cycles  (Ct) of 40 or above is not clear; (ii) there is no comparison using COV2 antigen detection; (iii) several statements do not seem appropriate ie beginning of the discussion section should include the fact that the  extensive COV2 testing is essential for monitoring viral infection in the human population and a prerequisite for investigating how the virus spreads.

We look forward to receiving your revised manuscript.

Kind regards,

Jean-Luc EPH Darlix, MG, Ph.D.

Academic Editor

PLOS ONE

---

## [Author Response · Author response to Decision Letter 1]

18 Jan 2021

Swab pooling: a new method for large-scale RT-qPCR screening of SARS-CoV-2 avoiding sample dilution (PONE-D-20-32256R1)

Rebuttal letter:

1. The reason for using amplification cycles (Ct) of 40 or above is not clear.

We have modified the text accordingly to clarify this point. We included more detailed information to consider a sample positive for SARS-CoV-2, which states as follows:

“For pool samples, detecting at least one gene with Cq value lower than 40 was enough for pool opening and subsequent individual testing; for individual samples, detecting both genes with Cq lower than 40 was required to consider it positive for SARS-CoV-2".

This was added to the methods section in lines 121-127. 

Three individual samples had divergent results (one gene with Cq lower than 40 and another with Cq slightly higher than 40), but it was not possible to collect new samples. For precaution, the patients were treated as positive and, hence, their results were kept in the analysis. This was detailed in the rewritten lines 203-211.

2. There is no comparison using COV2 antigen detection.

We understand the interest in antigen tests due to the benefits of point-of-care diagnosis. However, comparisons involving antigen testing go beyond the scope of this work for several reasons:

A. Antigen testing has lower sensitivity than RT-qPCR: while antigen testing has become popular, it is recommended that negative results should be confirmed by RT-qPCR due to inferior sensitivity (1). Even though it is suggested that antigen tests are sensitive enough to detect active infection only, there is substantial uncertainty regarding its actual sensitivity, with reported values ranging from 80% to 98% under varying validation conditions (2-4). Most works assessing antigen tests used RT-qPCR as a unique reference, as we do here. Future works comparing multiple methods should be useful, but beyond our current operational capacity and scope.

B. Antigen testing and swab pooling target different populations: while swab pooling, as any other pooling strategy, targets low prevalence populations, antigen testing has been recommended to target symptomatic individuals (1,4). In fact, a recent assessment comparing antigen testing to standard RT-qPCR found a sensitivity of 80% (32 of 40) among symptomatic individuals but of only 41% (7 of 17) for asymptomatic individuals (3). Even though this matter certainly requires further investigation, there is not enough evidence to support the widespread adoption of rapid antigen tests as a substitute of molecular testing for the screening of asymptomatic people and other low prevalence populations. Both the FDA and Brazil’s National Health Surveillance Agency (ANVISA) recommend that negative antigen test results should be subsequently confirmed by RT-qPCR, which may further delay the process of ruling out infection (1,4). The use of antigen tests for screening is suggested only if molecular testing is not feasible or if turnaround times are too long.

C. Our aim is solely to propose a reliable and efficient alternative for large-scale screening in low prevalence populations: massive molecular testing for SARS-CoV-2 places logistic, operational, and financial challenges. The more alternatives available the higher our ability to address issues due to insufficient testing. Here, our proposal improves on sample pooling, a strategy successfully implemented worldwide (5-6). 

Finally, works addressing different variations of pool testing continue to be published without any comparison to antigen testing (5-6). The value of our contribution lies in the improvement over the two main drawbacks from sample pooling: sample dilution and increased laboratory workload. Our method has allowed reliable screening of over 20 thousand individuals who would not have been tested by either individual RT-qPCR or antigen testing. 

3. Several statements do not seem appropriate i.e. the beginning of the discussion section should include the fact that the extensive COV2 testing is essential for monitoring viral infection in the human population and a prerequisite for investigating how the virus spreads.

 While we are not sure which statements the editor refers to, we have reviewed our discussion section and included the suggested observations (lines 205-206), in addition to minor modifications along the manuscript.

REFERENCES

1. FDA. 2020. A Closer Look at Coronavirus Disease 2019 (COVID-19) Diagnostic Testing. Available at: https://www.fda.gov/media/143737/download. Accessed: 15/01/2021.

2. https://doi.org/10.1016/S0140-6736(20)32635-0

3. http://dx.doi.org/10.15585/mmwr.mm695152a3

4. ANVISA. 2021. NOTA TÉCNICA Nº 7/2021. Available at: https://www.gov.br/anvisa/pt-br/centraisdeconteudo/publicacoes/servicosdesaude/notas-tecnicas/nota-tecnica-no-7-de-2021.pdf. Accessed: 15/01/2021.

5. https://www.thelancet.com/journals/laninf/article/PIIS1473-3099(20)30362-5/fulltext

https://www.nature.com/articles/s41586-020-2885-5

---

## [Editor Report · Decision Letter 2]

21 Jan 2021

Swab pooling: a new method for large-scale RT-qPCR screening of SARS-CoV-2 avoiding sample dilution

PONE-D-20-32256R2

Dear Dr. de Oliveira,

We’re pleased to inform you that your manuscript has been judged scientifically suitable for publication and will be formally accepted for publication once it meets all outstanding technical requirements.

Kind regards,

Jean-Luc EPH Darlix, MG, Ph.D.

Academic Editor

PLOS ONE
---

## [Editor Report · Acceptance letter]

26 Jan 2021

PONE-D-20-32256R2 

Swab pooling: a new method for large-scale RT-qPCR screening of SARS-CoV-2 avoiding sample dilution 

Dear Dr. Oliveira:

I'm pleased to inform you that your manuscript has been deemed suitable for publication in PLOS ONE. Congratulations! Your manuscript is now with our production department. 

Kind regards, 

on behalf of

Professor Jean-Luc EPH Darlix 

Academic Editor

PLOS ONE